# Dynamic Analysis and Simulation of the Feasibility and Stability of Innovative Carbon Emission Reduction Projects Entering the Carbon-Trading Market

Haotong Jiang , Liuyang Yao , Xueru Bai and Hua Li *

College of Economics and Management, Northwest A&F University, Yangling 712100, China
* Correspondence: lihua7485@163.com; Tel.: +86-133-6393-6398

**Abstract:** Designing green agricultural production projects as CER projects is attractive, as it can play a role in promoting the spread of green production technologies and reducing carbon emissions based on market-based compensation. This research constructed a generic analytical framework using evolutionary game methods to evaluate the feasibility and stability of innovative CER projects by numerical analysis or simulation. Finally, two complex scenarios were analysed using simulations based on the framework. The simulation analysis results show that when the profits of VER and CER projects are close, the government's direct intervention will lead to instability in market development, and the development of VER projects should be focused on. Government subsidies to promote the agricultural sector to participate in CER construction can be gradually reduced with the development of the market. When the reduction speed is slow enough, the effectiveness of subsidies will not be affected. The framework will be helpful to evaluate the feasibility and effectiveness of CER mechanism innovation and development, and to formulate more targeted policies to promote the popularization of green agricultural production technologies.

**Keywords:** clean development mechanism; evolutionary game; carbon markets; sustainable development; government regulation; behaviour; simulation

## 1. Introduction

Agriculture contributes 10–20% to global carbon emissions [1,2]; thus, low-carbon agricultural technology and production methods are important for carbon emission reduction and sustainable development [3,4]. However, farmers may be hindered by a short-term reduction in production efficiency, increased input costs [5–7], and time and learning costs [8,9]. Moreover, externalities of agriculture have long-term effects, yet farmers cannot obtain economic returns for them [10,11].

The carbon trading under the clean development mechanism (CDM) is an effective strategy to mitigate climate change. It is a market-based tool for trading greenhouse gas (GHG) emission rights [12]. Carbon trading markets use penalty and incentive systems to reduce emissions and encourage the adoption of low-carbon technologies [13,14]. Some regional clean development mechanisms also play a similar role, such as CCER (China certified emission reduction) [15,16]. Carbon-trading systems bring external benefits in the form of direct profits through market mechanisms [17]. Examples from Malaysia and Mozambique show that carbon-trading projects can reduce emissions and increase enthusiasm for green production technologies [18,19].

The certified emission reduction (CER) serves as a carbon-trading tool generated under the clean development mechanism, allowing companies without carbon quotas to offset their emissions by purchasing certified emission reductions. Clean development mechanism regulations provide incentives for sustainable development outcomes, such as poverty alleviation, job creation, and capital mobilization, by allowing the design of reductions in agricultural technologies and methods as CERs [20–24]. However, CER

trading faces various challenges and difficulties in practice. Evaluating the feasibility and sustainability of CDM projects is a crucial yet complicated issue. Currently, commonly used assessment methods mainly include cost-benefit analysis from a single-subject perspective [25–28], multi-criteria decision analysis for admission feasibility assessment [29,30] and life cycle analysis [31–34]. These methods have their own advantages and disadvantages, as well as some common limitations. For instance, they may ignore implicit or non-market costs or benefits, such as technological and regulatory barriers related to CDM projects in certain sectors [35,36]. Additionally, these methods may only consider the perspectives and objectives of individual subjects or stakeholders, such as governments, corporations, or project investors, without fully considering their interactions [37–39].

To address these limitations, we propose a new assessment method that uses evolutionary game theory to analyse behaviour selection and strategy changes among different subjects in CDM projects. Evolutionary game theory is a mathematical model that combines evolutionary biology and game theory to study the formation of stable strategies or equilibrium states of rational or bounded rational individuals in repeated games with adaptive and learning capabilities. The evolutionary game method is widely used in the feasibility study of energy utilization, policy governance, commercial projects, management models, and other fields [40–43].

Our approach using evolutionary game theory to evaluate the feasibility and sustainability of CDM projects includes several improvements compared to existing methods.

Firstly, we recognize that the carbon-trading market comprises carbon credit producers, consumers, and market order maintainers. Therefore, we consider direct participants such as governments, enterprises, and farmers, which allows for a more comprehensive analysis of the game relationship between stakeholders. Previous research primarily focused on the game between two direct participants, such as the government and CER investors for project construction [44,45], or the government and carbon emission enterprises for rent-seeking and illegal behaviour [46,47]. In some cases, indirect participants such as intermediaries or social groups were included [48–50]. By selecting direct participants, our approach improves the accuracy of measuring feasibility and stability.

Secondly, we have revised the types and numbers of model parameters based on the latest research results, and considered the changes of costs and benefits in the process of market development to better reflect the actual situation. For example, we included transaction costs of CER products [27,51], publicity benefits obtained by carbon emission enterprises when purchasing CER [52,53] and political interests [54].

Finally, compared with evaluation methods that consider a single subject cost, the application of evolutionary game theory to project evaluation is an innovative method that enhances the effectiveness of the evaluation results.

This paper utilizes a framework to analyse various issues pertaining to low-carbon agricultural technology, carbon-trading markets, and government subsidies. Specifically, we examine the feasibility of using low-carbon agricultural technology as a certified emission reduction (CER) to promote economic benefits in the agricultural sector. Additionally, we assess the stability of the status of low-carbon agricultural technology in the carbon-trading market, as well as the effect of profit gaps and government subsidies on evolutionary equilibrium strategy through numerical simulation. Based on these conclusions, recommendations are provided to the government regarding the construction of a carbon-trading platform and setting carbon-trading targets.

The remainder of this paper is organized as follows: Section 2 details the construction of the evolutionary game model employed in this study, including the stakeholder relationships, model parameters, and payoff matrix. Section 3 presents the results of our study, including the theoretical analysis process and the equilibrium state stability and evolutionary direction results, as well as the simulation process and results. Finally, Sections 4 and 5 discuss the research results and offer suggestions for policy and decision-making.

## 2. Materials and Methods

### 2.1. Logical Relationship of Evolutionary Game Model

During the establishment of a carbon emission reduction (CER)-trading market, the government, agricultural departments, and pollution emission enterprises engage in interactions regarding carbon credit trading and pollution regulation. The underlying logic of the three-party evolutionary game, which is constructed in this study, is illustrated in Figure 1.

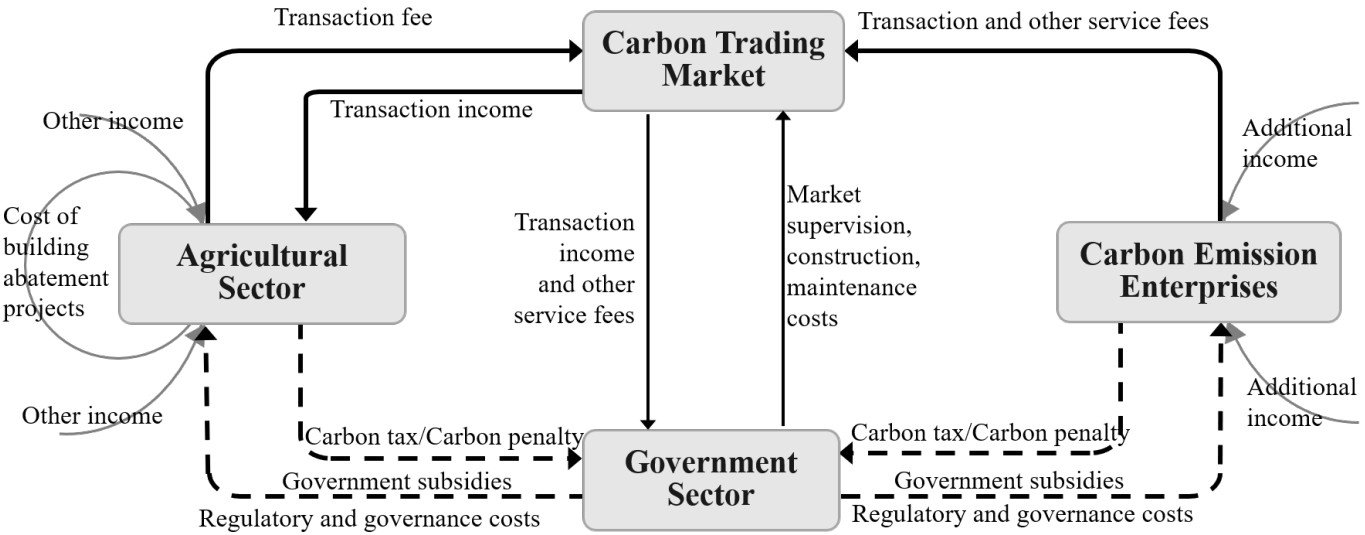

**Figure 1.** The logical relationship diagram of evolutionary game subject.

### 2.2. Assumptions of the Model

To assess the feasibility and stability of CERs, we proposed key variables and assumptions that underpin the model. We describe some of these variables below, while Table 1 provides further explanations.

**Assumption 1.** *The evolutionary game model includes three participants. The agricultural sector (Participant 1) comprises agricultural enterprises, collectives, or individuals that engage in agricultural production and may invest in agricultural reduction projects. The government sector (Participant 2) consists of state organs responsible for maintaining market order and incentivizing the establishment and implementation of CERs. The carbon-emitting enterprise (Participant 3) represents the main $CO_2$-emitting enterprises, subject to government and public opinion supervision for air pollution.*

**Assumption 2.** *The agricultural sector's strategy space, $\alpha = (\alpha 1, \alpha 2)$, includes constructing emission reduction projects and not constructing them, with a probability of $x \in [0, 1]$ and $(1 - x)$ being the probability of choosing $\alpha 2$. The government sector's strategy space, $\beta = (\beta 1, \beta 2)$, includes constructing certification systems and carbon-trading platforms and not constructing them, with a probability of $y \in [0, 1]$ and $(1 - y)$ being the probability of choosing $\beta 2$. The carbon-emitting companies' strategy space, $\gamma = (\gamma 1, \gamma 2)$, includes purchasing agricultural CERs and not purchasing them, with a probability of $z \in [0, 1]$ and $(1 - z)$ being the probability of choosing $\gamma 2$.*

**Assumption 3.** *Regardless of whether the government constructs a certification system and carbon-trading platform, the trading of reduction projects may still occur in the carbon-trading market [55–57]. Take China as an example, when the project cannot be developed and sold in accordance with the requirements of the United Nations Framework for Climate Change CDM Executive Committee (EB) or the National Development and Reform Commission (NDRC) for CDM projects due to some reasons, the project traded is called VER (voluntary emission reduction) [58]. Therefore,*

*for convenience, when both the agricultural sector and the carbon-emitting enterprise actively participate in the construction and purchase of reduction projects, the carbon-trading market (regardless of CER market or VER market) is referred to as an active market; when only one participant of the agricultural sector or the carbon-emitting enterprise actively participates, the carbon-trading market is referred to as a neutral market; when neither the agricultural sector nor the carbon-emitting enterprise construct or purchase reduction projects, the carbon-trading market is referred to as a negative market. The attitude of the market will affect the utility composition of each participant.*

**Assumption 4.** *When the government supports the construction of a certification system and carbon-trading platform, the construction of reduction projects by the agricultural sector can sell CERs on the carbon market and obtain revenue $I_C^1$. When the government does not support the construction of a certification system and carbon-trading platform, the agricultural sector can sell VERs through the voluntary market, obtaining revenue $I_C^2$. The voluntary market is smaller in scale and has lower regulatory and transaction costs, resulting in lower carbon credit prices [59]. Therefore, it is assumed that the revenue obtained by the agricultural sector through the trade of verified reduction projects is higher, i.e., $I_C^1 > I_C^2$.*

**Assumption 5.** *In an active verified reduction project market the cost of carbon emission supervision by the government will be effectively reduced, denoted as $C_G$. In order to achieve environmental goals, the government, out of prudence, cannot lower supervision intensity; therefore, the cost of supervision remains at a certain level and is difficult to significantly reduce. Therefore, $C_G < \widetilde{C_G}$.*

**Assumption 6.** *The construction of emission reduction projects by the agricultural sector can improve the production and living environment [60], which may have the effect of increasing agricultural output and improving the quality of life. This part of the benefit is recorded as additional revenue $I_A$, with $I_A < I_C$. Carbon-emitting companies can achieve additional benefits, such as social reputation or promotional effects, by offsetting their emissions through the purchase of carbon credit generated by emission reduction projects [51,52], denoted as $I_A^2$.*

**Table 1.** Variable symbols and descriptions.

| Stakeholders | Parameters | Descriptions |
|---|---|---|
| Agricultural Sector | $C_R$ | Cost of constructing emission reduction projects (including asset investment, loss of production reduction due to environmental protection, etc.) |
| | $I_C^1$ | Revenue from trading CER projects in the carbon-trading market |
| | $I_C^2$ | Revenue from trading VERs projects in the voluntary trading market |
| | $I_A$ | Additional income |
| | $S_C^1$ | Government subsidies for the construction of CER projects |
| | $P_1$ | Fines or taxes levied by the government for environmental pollution on the agricultural sector |
| | $R$ | Reduction in crop yield due to environmental issue |
| Government Sector | $E_U$ | Environmental benefits of an active emission reduction project trading market |
| | $E_U^*$ | The environmental benefits brought about by the trading market for emission reduction projects are only possible when the construction body actively participates |
| | $\widetilde{E_U}$ | Environmental benefits of an unregulated voluntary emissions trading market |
| | $\widetilde{E_U^*}$ | The environmental income of an unregulated voluntary emissions trading market when only the agricultural sector actively participates |
| | $I_T$ | Revenue generated by CER project transactions and other service fee revenue |
| | $C_V$ | Cost of supervising the implementation of certification projects or certification institutions |
| | $C_S^1$ | The cost of financial support for the agricultural sector constructing CER projects in a carbon-trading market |
| | $C_S^2$ | The cost of financial support for the carbon emitting-enterprises purchasing CERs in a carbon-trading market |

**Table 1.** *Cont.*

| Stakeholders | Parameters | Descriptions |
|---|---|---|
| | $C_M$ | The cost of building a certification system and carbon-trading platform |
| | $C_G$ | The cost of government regulation of carbon emissions in an active CER market |
| | $\widetilde{C_G}$ | The cost of government regulation of carbon emissions when there is no active market for trading certified emission reduction projects |
| | $I_P$ | Revenue from government taxes or penalties on carbon emissions when there is no active CER market |
| Carbon-Emitting Companies | $S_C^2$ | Government subsidies for carbon-emitting enterprises to purchase CER |
| | $I_A^2$ | Additional income for enterprises participating in carbon trading, including social reputation, publicity effects, etc. |
| | $C_E^1$ | Cost of purchasing carbon credits for agricultural CER projects and other related costs |
| | $C_E^2$ | Cost of purchasing carbon credits for other CER projects and other related costs |
| | $C_E^3$ | Cost of purchasing carbon credits for VER projects and other related costs |
| | $P_2$ | Government taxes or fines on carbon-emitting companies |

*2.3. Model Construction*

In order to construct an evolutionary game framework for assessing the feasibility and stability of innovative carbon-trading products, it is necessary to not only consider the respective benefit and cost parameters of the stakeholders, but also to take into account the interaction between their decisions and the effects of the interaction on profit and cost. Based on the above assumptions and variables, the three-party evolutionary game model is constructed, as shown in the Table 2.

**Table 2.** Payoff matrix of evolutionary game analysis.

| | | Constructing | Not Constructing |
|---|---|---|---|
| | | **Government Sector $y$** | **Government Sector $(1-y)$** |
| Active market | Agricultural sector $x$ Carbon-emitting companies $z$ | $S_C^1 + I_C^1 + I_A - C_R,$ $E_U + I_T - C_V - C_M - C_S^1 - C_S^2 - C_G,$ $S_C^2 + I_A^2 - C_E^1,$ | $I_C^2 + I_A - C_R,$ $\widetilde{E}_U - \widetilde{C_G},$ $I_A^2 - C_E^3 - P_2,$ |
| Neutral market | Agricultural sector $x$ Carbon-emitting companies $(1-z)$ | $S_C^1 + I_A - C_R$ $E_U^* + I_P - C_V - C_M - C_S^1 - \widetilde{C_G}$ $-P_2$ | $I_A - C_R$ $\widetilde{E}_U^* + I_P - \widetilde{C_G}$ $-P_2$ |
| | Agricultural sector $(1-x)$ Carbon emitting companies $z$ | $-R - P_1$ $I_P - C_M - C_S^2 - \widetilde{C_G}$ $S_C^2 + I_A^2 - C_E^2$ | $-P_1 - R$ $I_P - \widetilde{C_G}$ $I_A^2 - C_E^3 - P_2$ |
| Negative market | Agricultural sector $(1-x)$ Carbon-emitting companies $(1-z)$ | $-R - P_1$ $I_P - C_M - \widetilde{C_G}$ $-P_2$ | $-P_1 - R$ $I_P - \widetilde{C_G}$ $-P_2$ |

**3. Results**

*3.1. Strategy Stability Analysis*

3.1.1. Strategic Stability Analysis of Agricultural Sector

The expected income and average income of the agricultural sector for the choice of constructing emission reduction projects or not are shown in Formula (1):

$$\begin{cases} E_{11} = I_A^1 - C_R + I_C^2 z + S_C^1 y - I_C^2 yz \\ E_{12} = y(P_1 + R)(z - 1) - yz(P_1 + R) - (P_1 + R)(y - 1)(z - 1) + z(P_1 + R)(y - 1) \\ \overline{E}_1 = x E_{11} + (1 - x) E_{12} \end{cases} \tag{1}$$

The dynamic replicator equation for the agricultural sector's strategy choice is:

$$\begin{aligned}
F(x) &= \frac{\mathrm{d}x}{\mathrm{d}t} \\
&= x\left(E_{11} - \overline{E}_1\right) \\
&= -x\,(x-1)\left(I_A^1 - C_R + P_1 + R + I_C^2\,z + S_C^1\,y - I_C^2\,y\,z\right)
\end{aligned} \tag{2}$$

The first derivative of $x$ and the set $G(z)$ are, respectively:

$$\begin{aligned}
\frac{\mathrm{d}F(x)}{\mathrm{d}x} &= -(2x-1)\left(I_A^1 - C_R + P_1 + R + I_C^2\,z + S_C^1\,y - I_C^2\,y\,z\right) \\
&= -x(x-1)G(z).
\end{aligned} \tag{3}$$

when $z = z^* = \frac{-I_A^1 - C_R + P_1 + R + S_C^1\,y}{I_C^2\,(1-y)}$, $G(z) \equiv 0$. According to the principle of the stability of differential equations, the stable state of the agricultural sector's choice to construct emission reduction projects must satisfy $F(x) = 0$ and $\frac{\mathrm{d}F(x)}{\mathrm{d}x} < 0$.

Since $\frac{\partial G(z)}{\partial z} > 0$, $G(z)$ is an increasing function of $z$. Therefore, when $z > z^*$, $G(z) > 0$, and the evolutionary stable strategy (ESS) of the equation is $x = 1$; conversely, when $z < z^*$, $G(z) > 0$, $x = 0$ is the ESS of the equation.

As shown in the Figure 2, the probability of the agricultural sector not building emission reduction projects is $V_{A1}$, and the probability of building emission reduction projects is $V_{A2}$. This expression is shown in Formulas (A1) and (A2).

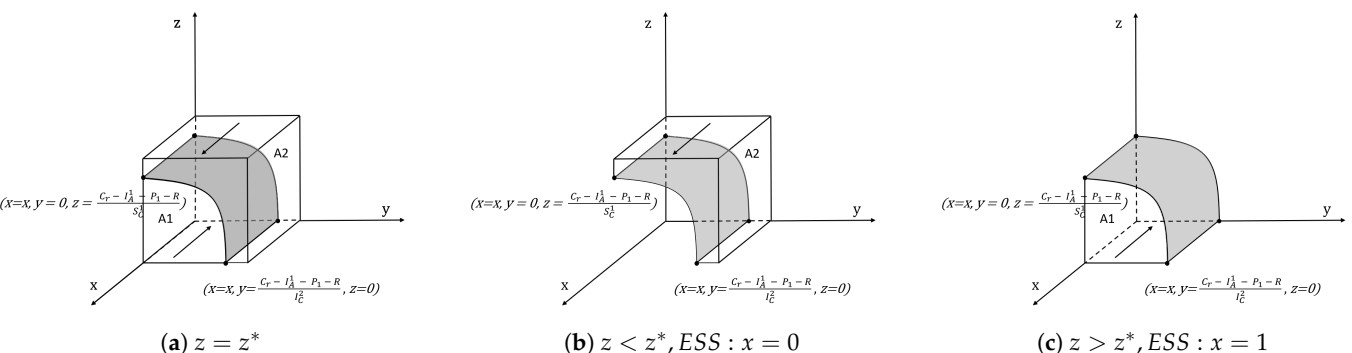

**(a)** $z = z^*$  **(b)** $z < z^*, ESS: x = 0$  **(c)** $z > z^*, ESS: x = 1$

**Figure 2.** Phase diagram of agricultural sector evolution.

3.1.2. Strategic Stability Analysis of Government Sector

The expected average returns of the government sector's decision to establish a certification system and a carbon-trading platform are, respectively, represented in Formula (4):

$$\left\{\begin{aligned}
E_{21} &= x\,(z-1)\left(\widetilde{C_G} + C_M + C_S + C_V - E_U^* - I_P\right) \\
&\quad - x\,z\left(C_G + C_M + C_S^1 + C_S^2 + C_V - E_U - I_T\right) \\
&\quad + z\,(x-1)\left(\widetilde{C_G} + C_M + C_S^2 - I_P\right) - (x-1)\,(z-1)\left(\widetilde{C_G} + C_M - I_P\right). \\
E_{22} &= I_P - \widetilde{C_G} + \widetilde{E}_U^*\,x + \widetilde{E}_U\,x\,z - \widetilde{E}_U^*\,x\,z - I_P\,x\,z. \\
\overline{E}_2 &= y\,E_{21} + (1-y)\,E_{22}.
\end{aligned}\right. \tag{4}$$

The replicator dynamic equation for the government sector's strategy choice is as follows:

$$\begin{aligned}
F(y) &= \frac{\mathrm{d}y}{\mathrm{d}t} \\
&= y\left(E_{21} - \overline{E}_2\right).
\end{aligned} \tag{5}$$

The first derivative of $y$ and set $J(x)$ are shown in Formula (8) (the proof is given in the Appendix A):

$$\begin{aligned}
\frac{\mathrm{d}F(y)}{\mathrm{d}y} &= (2y-1)\left(C_M + C_S^1 x + C_V x + C_S^2 z + \widetilde{E}_U^* x - E_U^* x + C_G xz - \widetilde{C_G} xz \right.\\
&\quad \left. - E_U xz + \widetilde{E}_U xz - \widetilde{E}_U^* xz + E_U^* xz - I_T xz \right)\\
&= (2y-1)J(x).
\end{aligned} \tag{6}$$

when $x = x^* = -(C_M + C_S^2 z)/(C_S^1 + C_V + \widetilde{E}_U^* - E_U^* + C_G z - \widetilde{C_G} z - E_U z + \widetilde{E}_U z - \widetilde{E}_U^* z + E_U^* z - I_T z)$, $J(x) \equiv 0$. $J(x)$ is always greater than 0 ($1 > z > 0, 1 > x > 0$) (the proof is given in the Appendix B). According to the principle of the stability of differential equations, the stable state of the government's decision to construct a certification system and a carbon-trading platform must satisfy $F(y) = 0$ and $\frac{\mathrm{d}F(y)}{\mathrm{d}y} < 0$. Therefore, when the net loss of the certification system and carbon-trading platform is present, $J(x) > 0$, and $y = 0$ is always the government's evolutionary stable strategy.

It is worth noting that as time progresses and government assessment methods change, the CERs trading system may result in net benefits for the government. In this case, $\frac{\mathrm{d}J(x)}{\mathrm{d}x} < 0$; therefore, $J(x)$ is a decreasing function of $x$. When $x > x^*$, $J(x) < 0$, and $dF(y)/dy|_{y=1} < 0$, $y = 1$ represent the evolutionary stable strategy of the equation, while $y = 0$ represents the evolutionary stable strategy.

The three-party evolutionary game with net profit is shown in Figure 3. The probability of the government not constructing reduction projects is $V_{B1}$ and the probability of constructing such projects is $V_{B2}$. As shown in the figure, under certain conditions, the analytical expression can be written as Formulas (A3) and (A4).

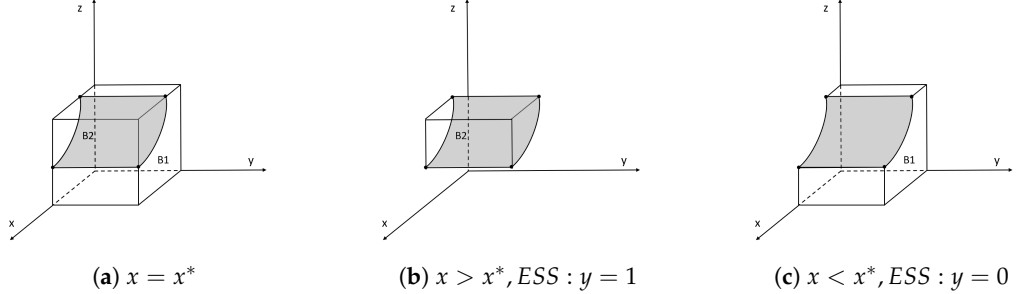

(**a**) $x = x^*$      (**b**) $x > x^*, ESS : y = 1$      (**c**) $x < x^*, ESS : y = 0$

**Figure 3.** Phase diagram of government sector evolution.

### 3.1.3. Strategic Stability Analysis of Carbon-Emitting Enterprises

The expected income and average income of carbon-emitting enterprises depending on whether to build emission reduction projects are:

$$\begin{cases}
E_{31} = I_A^2 - C_E^3 - P_2 - C_E^2 y + C_E^3 y + P_2 y + S_C^2 y - C_E^1 xy + C_E^2 xy.\\
E_{32} = P_2 x(y-1) - P_2 xy - P_2(x-1)(y-1) + P_2 y(x-1).\\
\overline{E}_3 = y E_{31} + (1-y) E_{32}.
\end{cases} \tag{7}$$

The replicator dynamic equation for the agricultural sector's strategy choice is as follows:

$$\begin{aligned}
F(z) &= \frac{\mathrm{d}z}{\mathrm{d}t}\\
&= z(E_{31} - \overline{E}_3)\\
&= -z(z-1)\left(I_A^2 - C_E^3 - C_E^2 y + C_E^3 y + P_2 y + S_C^2 y - C_E^1 xy + C_E^2 xy\right).
\end{aligned} \tag{8}$$

The first derivative of $z$ and the set $H(y)$ are, respectively:

$$\frac{dF(z)}{dz} = -(2z-1)\left(I_A^2 - C_E^3 - C_E^2 y + C_E^3 y + P_2 y + S_C^2 y - C_E^1 x y + C_E^2 x y\right)$$
$$= -(2z-1)H(y). \tag{9}$$

when $y = y^* = (C_E^3 - I_A^2)/(C_E^3 - C_E^2 + P_2 + S_C^2 - C_E^1 x + C_E^2 x)$, $H(y) \equiv 0$. According to the principle of the stability of differential equations, the stable state of the carbon-emitting enterprise's decision to purchase agricultural CER products must satisfy $F(z) = 0$ and $dF(z)/dz < 0$. Since $dH(y)/dy = C_E^3 - C_E^2 + P_2 + S_C^2 - C_E^1 x + C_E^2 x > 0$, $H(y)$ is an increasing function of $y$. Therefore, when $y > y^*$, $H(y) > 0$, and the ESS of the equation is $z = 1$; conversely, when $y < y^*$, $H(y) < 0$, and $z = 0$ is the ESS of the equation.

As shown in Figure 4, the probability of carbon-emitting enterprises not building emission reduction projects is $V_{C1}$, and the probability of building emission reduction projects is $V_{C2}$. Under certain conditions, the analytical expression can be written as Formulas (A5) and (A6).

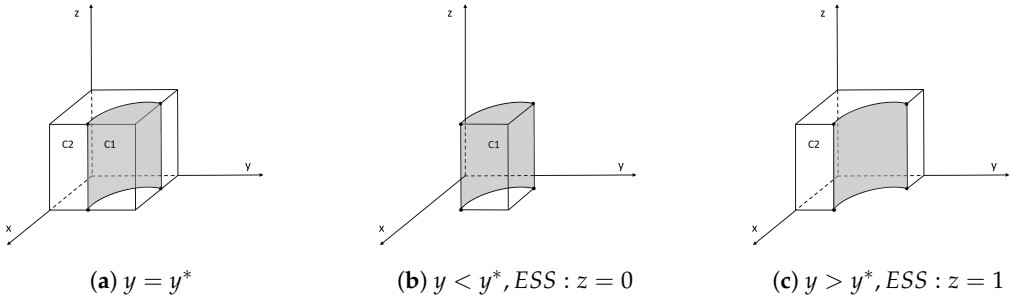

(**a**) $y = y^*$　　　　(**b**) $y < y^*, ESS : z = 0$　　　　(**c**) $y > y^*, ESS : z = 1$

**Figure 4.** Phase diagram of a carbon-emitting enterprise.

### 3.2. Analysis of System Equilibrium Stability

Based on $F(x) = 0$, $F(y) = 0$, and $F(z) = 0$, the system equilibrium points are $E1(0,0,0)$, $E2(1,0,0)$, $E3(0,1,0)$, $E4(0,0,1)$, $E5(1,1,0)$, $E6(1,0,1)$, $E7(0,1,1)$, and $E8(1,1,1)$. The stability of the equilibrium points can be analysed by analysing the Jacobian matrix of the three-party evolutionary game system: if all the eigenvalues of the Jacobian matrix have negative real parts, the equilibrium point is a stable point; if at least one eigenvalue has a positive real part, the equilibrium point is an unstable point; if the Jacobian matrix has eigenvalues with zero real parts except for the eigenvalues with negative real parts, the equilibrium point is in a critical state and the stability cannot be determined by the eigenvalue sign. The Jacobian matrix of the three-party evolutionary game system is:

$$J = \begin{bmatrix} \frac{dF(x)}{dx} & \frac{dF(x)}{dy} & \frac{dF(x)}{dz} \\ \frac{dF(y)}{dx} & \frac{dF(y)}{dy} & \frac{dF(y)}{dz} \\ \frac{dF(z)}{dx} & \frac{dF(z)}{dy} & \frac{dF(z)}{dz} \end{bmatrix} \tag{10}$$

The stability of the equilibrium point of a non-linear system can be determined using the Lyapunov first method, with the following criteria:

1. When all the eigenvalues of the Jacobian matrix $J$ have negative real parts, the system $A$ is asymptotically stable and the equilibrium state $E$ is also asymptotically stable, meaning that the system converges asymptotically.
2. When at least one eigenvalue of $J$ has a positive real part, the system $A$ is unstable and the equilibrium state $E$ is also unstable.
3. When none of the eigenvalues of $J$ have positive real parts, but at least one eigenvalue has a zero real part, then the stability of the system and equilibrium state $E$ must be verified using other methods.

According to the Lyapunov first method, the stability of the equilibrium point can be determined as shown in the Table 3:

**Table 3.** Stability of equilibrium.

| Equilibrium | Eigenvalue | | | | Stability |
|---|---|---|---|---|---|
| | $\lambda_1$ | $\lambda_2$ | $\lambda_3$ | Real Part Symbol | |
| $E_1(0,0,0)$ | $I_A - C_R + P^1 + R$ | $-C_M$ | $I_A^2 - C_E^3$ | $(+,-,\times)$ | Instability point |
| $E_2(1,0,0)$ | $-I_A + C_R - P^1 - R$ | $-C_M + C_S^1 + C_V + \widetilde{E}_U^* - E_U^*$ | $-C_E^3 + I_A^2$ | $(-,\times,\times)$ | Uncertain |
| $E_3(0,1,0)$ | $I_A - C_R + P^1 + R + S_c^1$ | $C_M$ | $I_A^2 - C_E^3 + P^2 + S_C^2$ | $(+,+,+)$ | Instability point |
| $E_4(0,0,1)$ | $I_A - C_R + I_C^2 + P^1 + R$ | $-C_M - C_s^2$ | $C_E^3 - I_A^2$ | $(+,-,\times)$ | Instability point |
| $E_5(1,1,0)$ | $-I_A - C_R + P^1 + R + S_C^1$ | $C_M + C_S^1 + C_V + \widetilde{E}_U^* - E_U^*$ | $I_A^2 - C_E^1 + P^2 + S_C^2$ | $(-,\times,+)$ | Instability point |
| $E_6(1,0,1)$ | $-I_A - C_R + I_c^2 + P^1 - R$ | $-(C_G + \widetilde{C}_G - C_M - C_s^1 - C_s^2 - C_v + E_U - \widetilde{E}_U + I_T)$ | $C_E^3 - I_A^2$ | $(\times,-,\times)$ | Uncertain |
| $E_7(0,1,1)$ | $I_A - C_R + P^1 + R + S_c^1$ | $C_M + C_s^2$ | $I_A^2 - C_E^2 + P^2 + S_C^2$ | $(\times,+,\times)$ | Instability point |
| $E_8(1,1,1)$ | $-I_A - C_R + P^1 + R + S_C^1$ | $C_G - \widetilde{C}_G + C_M + C_s^1 + C_s^2 + C_v - E_U + \widetilde{E}_U - I_T$ | $-I_A^2 + C_E^1 - P^2 - S_C^2$ | $(-,\times,-)$ | Uncertain |

The success of a CER project relies on the interaction between the government, agricultural sector and carbon-emitting enterprises. Analysis of the equilibrium points $E_2$, $E_3$ and $E_4$ shows that when only one party promotes low-carbon production methods, unstable and unsustainable market conditions occur. To promote green agricultural production methods, these three parties' mutual interests must be considered. This will help drive the development of the carbon-trading market.

Based on the assumption that farmers will spontaneously maintain the agricultural ecological environment, the analysis of the equilibrium point $E_6$ shows that when carbon-emitting companies gain additional benefits such as social prestige and advertising effects through reducing emissions, which are greater than their costs, the $E_6(1,0,1)$ is a stable strategy and companies and farmers will form a carbon-trading market without government supervision. This has been supported by many studies.

If we reject the assumption that farmers will spontaneously maintain the agricultural ecological environment, point $E_1$ will become the evolutionarily stable point, that is, this innovative computer program has no market feasibility. However, point $E_5$ shows that appropriate government subsidies can promote the efficiency of innovative CERs and $E_7$ point illustrates that agricultural CERs trading emissions is a necessary step to solve carbon pollution.

### 3.3. Simulation Analysis

Considering the influence of initial strategy selection and influence factors on the evolution results, we use Matlab 2021b to carry out numerical simulation and assign values to the model in combination with literature and reality. On the basis of meeting the assumptions of the model presented in Section 2.2, the impact of variable changes on the evolutionary game process and results are analysed. The simulation analysis is based on the following parameter settings: $C_R = 20, I_C^1 = 50, I_C^2 = 20, I_A^1 = 8, S_C^1 = 20, P_1 = 0, R = 5, E_U = 300,$ $E_U^* = 20, E_U^2 = 40, \widetilde{E}_U^* = 8, I_T = 20, C_V = 20, C_S = 20, C_S^2 = 10, C_M = 50, C_G^1 = 10,$ $\widetilde{C}_G = 300, I_P = 30, S_C^2 = 10, I_A^2 = 5, C_E^1 = 10, C_E^2 = 10, C_E^3 = 8, P_2 = 20.$

### 3.3.1. Effect of Construction Costs of CER Projects on Strategy Selection

The construction costs of agricultural emission reduction projects are key variables in the CER trading systems, including direct costs, indirect costs and opportunity costs, which have an important impact on the strategic selection of farmers, and thus on the decision making of governments and enterprises. We discuss the influence of construction cost factors on the evolutionary outcomes. The simulation results are shown in the Figures 5 and 6.

From the parameter allocation, it can be seen that for a certain emission reduction project, the construction cost is $C_R$, the CER trading income is $I_C^2$, and the subsidy provided by the government to the agricultural sector for the construction of such emission reduction projects is $S_C^1$. When the subsidy can cover all or part of the cost ($C_R = 20, C_R = 30$), due to the influence of additional income and environmental productivity improvement, the construction of emission reduction projects is a gradual and stable strategy for the agricultural sector. However, when the cost of such construction projects is close to or equal to its returns ($C_R = 40, C_R = 50$), even with government subsidies and additional income, the construction of emission reduction projects is not a gradual and stable strategy for the agricultural sector. We have made a robustness analysis of the $x, y,$ and $z$ starting conditions for $C_R = 40$ in Figure 7, and under any initial decision ratio, the system's stable solution is $(0, 0, 0)$, which confirms the unsustainability of the CER market under higher construction costs.

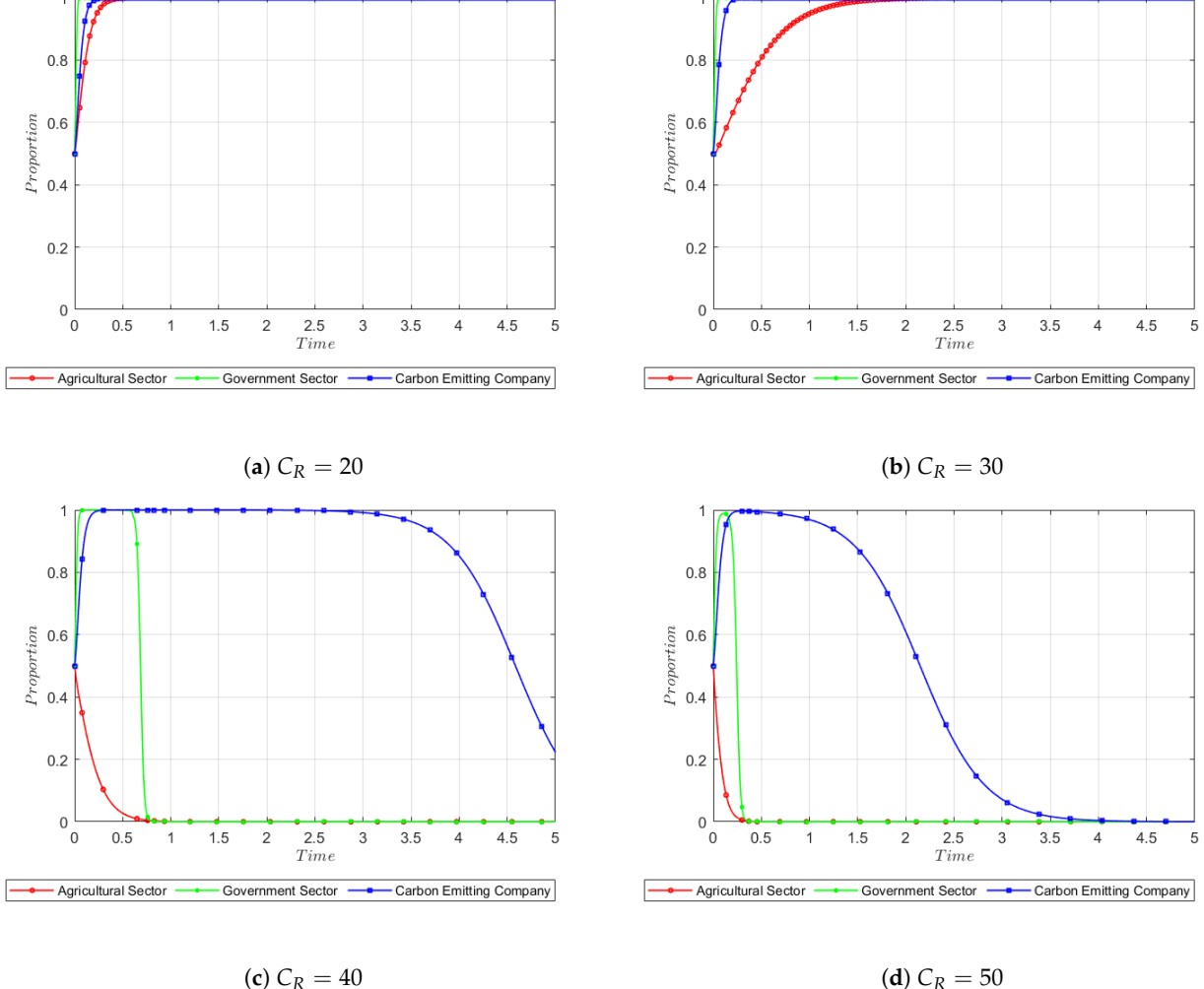

(**a**) $C_R = 20$

(**b**) $C_R = 30$

(**c**) $C_R = 40$

(**d**) $C_R = 50$

**Figure 5.** The effect of construction cost on the evolutionary strategies from a separation perspective.

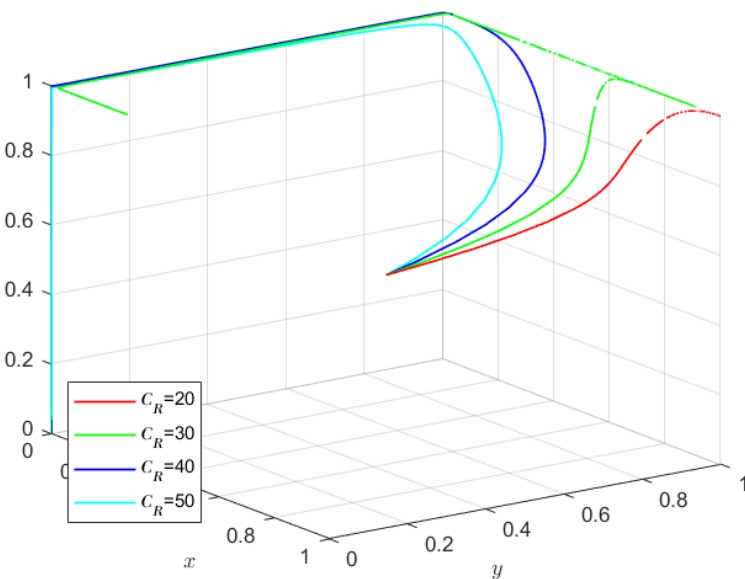

**Figure 6.** The effect of construction cost on the evolutionary strategies from a unified perspective.

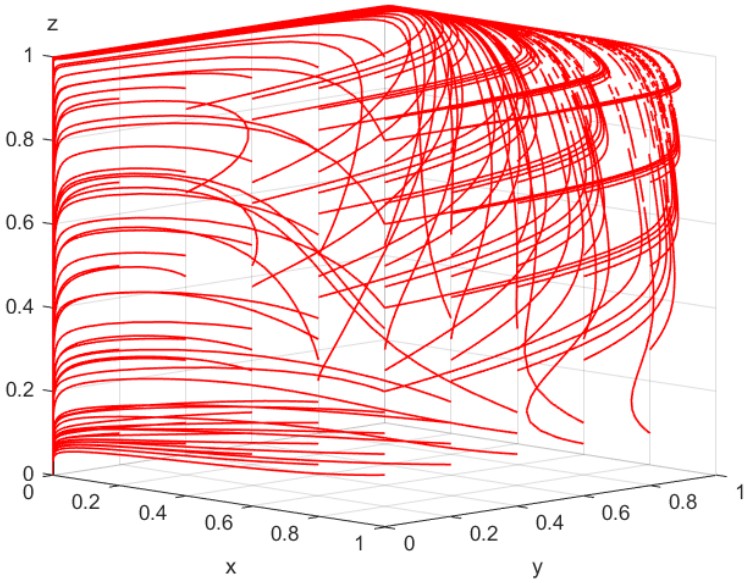

**Figure 7.** The effect of construction cost on the evolutionary strategies from a unified perspective.

### 3.3.2. Effect of Profit Gap between CER and VER on Strategy Selection

The difference in profits that the agricultural sector gain from the CER and VER trading is a key variable affecting whether the agricultural sector actively participates in CER trading with stringent standards. As the standards of VER projects are lower, some of the approval processes are eliminated and costs saved, leading to an increased success rate of development and lower transaction prices, whereas CER has stringent reviews, good environmental benefits, and higher transaction prices that are beneficial to increasing profits for the agricultural sector. Both have advantages and disadvantages. For the three parties involved in the emission reduction trading model, the profit gap between CER and VER is an important factor influencing their strategy decisions. Therefore, we discuss how disparity between CER and VER revenue affects the evolutionary outcome in Figure 8.

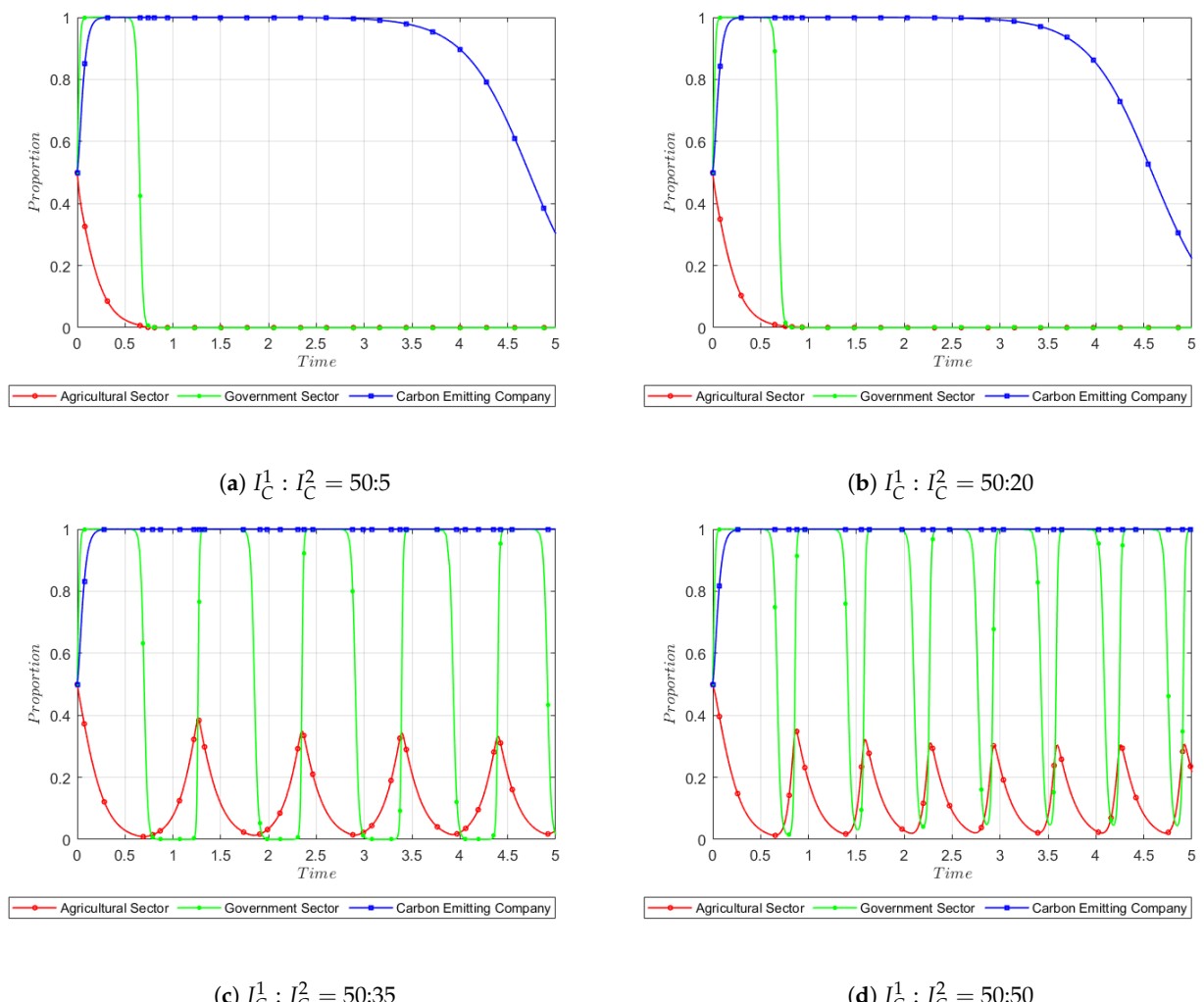

**(a)** $I_C^1 : I_C^2 = 50{:}5$　　　　　　　　　　　　　**(b)** $I_C^1 : I_C^2 = 50{:}20$

**(c)** $I_C^1 : I_C^2 = 50{:}35$　　　　　　　　　　　　　**(d)** $I_C^1 : I_C^2 = 50{:}50$

**Figure 8.** The effect of profit gap between CER and VER on the evolutionary strategies.

When the voluntary carbon-trading market generates high profits, the enthusiasm of the agricultural sector to participate will increase. At this time, the government may show willingness to participate in the construction of the carbon market, bringing higher transaction costs, more standardized construction costs (including the cost of achieving stricter emission reduction standards, etc.), causing market enthusiasm to decline, thus disrupting the process of market evolution promoting an unstable state of fluctuation. In Figure 9, we show two policy measures that the government may take to avoid the instability of the carbon-trading market: the government can choose to not intervene in market development (i.e., encourage the development of VER trading) or take appropriate pollution penalty measures for the agricultural sector.

### 3.3.3. Strategy Selection under Dynamic Subsidies

Government subsidies for farmers may decrease over time due to the gradual improvement of the market system and the path-dependence effect, thus the stability of the evolutionary strategy in the agricultural sector may experience certain changes. To illustrate this, we replaced the variable $S_C^1$ with an S-shaped decreasing variable expressed by a logistic function, $S_C^{1'} = S_C^1 / e^{\lambda t}$, where $\lambda$ represents the decay rate. We analyse how the rate of subsidy decay affects evolutionary outcomes, as illustrated in Figure 10 .

From the analysis, it can be seen that government subsidies reducing too quickly may increase the uncertainty of the market. When the government subsidies are reduced at a slower speed, the market can form spontaneous stability.

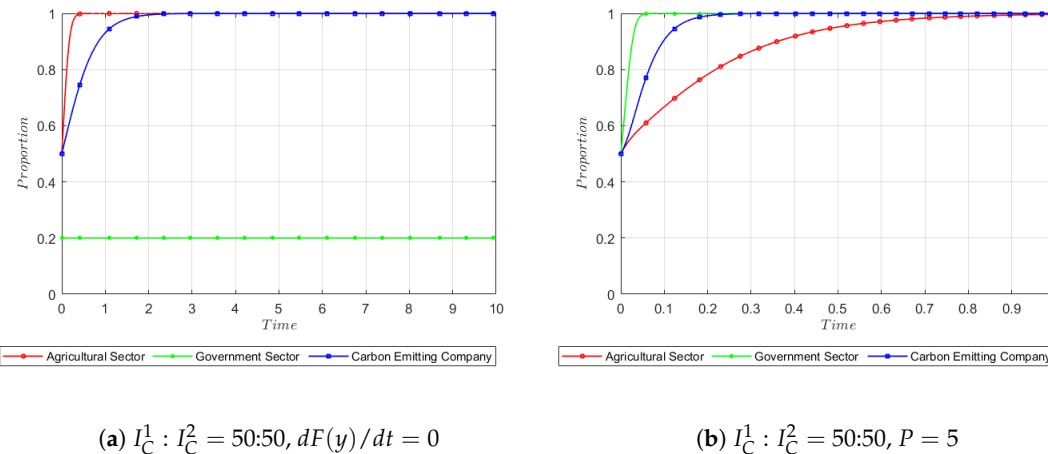

(**a**) $I_C^1 : I_C^2 = 50{:}50, dF(y)/dt = 0$          (**b**) $I_C^1 : I_C^2 = 50{:}50, P = 5$

**Figure 9.** Correction strategies for market instability.

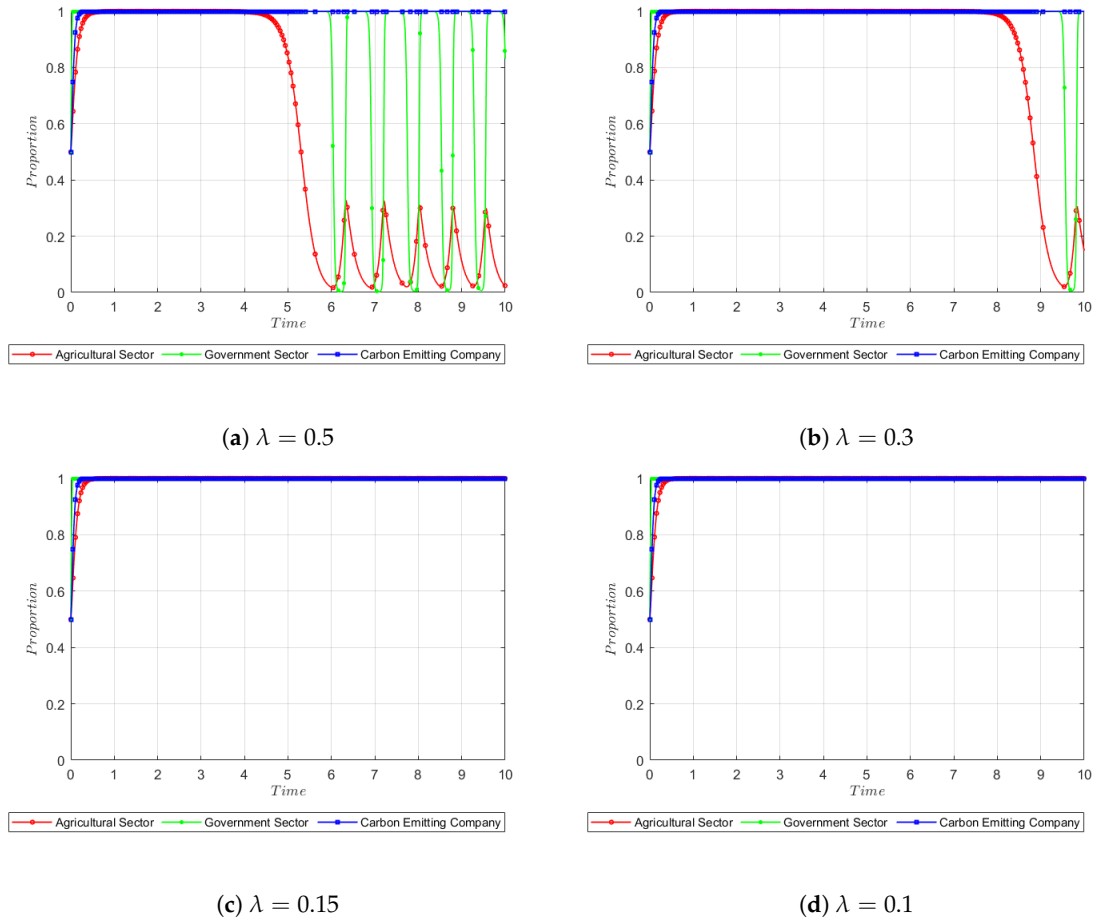

(**a**) $\lambda = 0.5$                        (**b**) $\lambda = 0.3$

(**c**) $\lambda = 0.15$                     (**d**) $\lambda = 0.1$

**Figure 10.** Th effect of dynamic subsidies on the evolutionary strategies.

## 4. Discussion

An agricultural production method that has a significant emission reduction effect can be developed into an innovative clean development mechanism (CDM) project. The establishment of innovative CDM projects will enhance the vitality of the carbon-trading market, stimulate the innovation and advancement of carbon-reduction technologies and methodologies, and incentivize active participation from all stakeholders through economic benefits. This study introduces a novel paradigm for feasibility analysis, which utilizes an evolutionary game framework to predict and evaluate through simulation. Further-

more, the impact of parameters on the feasibility assessment was investigated through simulation analysis.

Firstly, it is widely acknowledged in the literature that successful CER projects must have financial viability and sustainability in their business models [61–64]. To support this view, we conducted a simulation analysis by changing the initial conditions and construction cost parameters. Our research findings indicate that even with government subsidies, high costs make projects unfeasible and unsustainable. Furthermore, theoretical analysis of stable system solutions shows that projects with high regulatory costs are unsustainable for governments. Therefore, it is crucial for CER projects to have verifiable and validated emission reduction or a high emission reduction capacity to ensure sustained government support [65]. The experience of US renewable energy policies suggest that overly aggressive subsidy policies can lead to policy failures [66,67], while Japan's research shows that sustainable policy design is a key factor for policy success [68]. This result suggests that government policymakers should pay particular attention to regulatory costs and policy sustainability because regulatory costs and sustainability are often overlooked in the early stages of policy making [69].

Secondly, we explored the interaction between VER and CER projects by comparing their profits. The literature suggests that VER projects have real significance in increasing the income of low-income populations and reducing carbon emissions, and have the potential for rapid development [70,71]. Corbera (2009) argues that the CER mechanism is more suitable for small-scale projects [17], while Muller (2007) believes that projects with low costs and lower additional benefits are more attractive to investors [72,73]. Through simulation analysis, we identified suitable conditions and scenarios for CER projects to be converted into VER projects. When the profits of VER and CER are similar, or when VER profits are higher, the market tends to develop VER projects, but at this point, the system does not have a stable solution, and the market oscillates between CER and VER orientations. Promoting the development of VER projects can stabilize market evolution results. However, if the development of CER is politically necessary, it may be necessary to impose appropriate penalties on farmers' carbon emissions to ensure stable evolutionary results. As the voluntary carbon market as a whole has not yet found a way to be compatible with the new legal framework of the Paris Agreement in a credible and legitimate manner [53], appropriately regulating to limit the trend of CER projects towards VER projects may be a better choice. Therefore, the framework proposed in this study provides a theoretical test method for governments to develop development plans.

Thirdly, government subsidies play a critical role in promoting the development of CER trading. Many studies indicate that government subsidies play an important role in promoting CER projects through various methods [48,74,75]. We aimed to explore whether government subsidies must be sustainable. We introduced a government subsidy function that increases with time $t$ and decreases with an S-shaped curve. We then simulated the changes in equilibrium solutions under different government subsidy reduction rates. Chen (2018) used the framework of evolutionary game theory to demonstrate the effectiveness of another alternative dynamic government subsidy form [76]. The results show that the government can choose a reduction rate that balances subsidy needs and financial sustainability, and the effectiveness of subsidies depends on how the government manages the subsidy reduction process. The findings of this study suggest that sustainable and efficient government subsidies are essential for the successful development of CER projects.

## 5. Conclusions and Policy Suggestion

The evaluation framework built by this study is conducive to the formulation of more reasonable carbon-trading policies. The framework enables policymakers to use predicted parameter values to analyse the potential effects of innovative CER, or to adjust their own decision parameters based on the expected effects. The framework is conducive to the government taking appropriate measures to reduce policy costs, improve policy utility,

and achieve reasonable policy expectations. Based on the research results, we propose the following policy recommendations:

1.  In the CDM mechanism of agricultural green development, agricultural sector, government sector and carbon-emitting enterprises all play an important role. In the process of policy formulation, it is necessary to consider the benefits and losses of multiple stakeholders.
2.  Conducting extensive popular science publicly can help farmers realize the concept that agricultural pollution will reduce crop yield, protect the environment and economic interests, and even bring extra benefits, effectively reducing the cost of policy implementation and promote the enthusiasm of the agricultural department.
3.  CER projects should have a basic financial feasibility in terms of costs and benefits, otherwise, they may not be sustainable in the promotion process. By improving the level of technology, reducing the construction cost of CER projects can help to promote the adoption of low-carbon agricultural production technology and the development of carbon-trading markets.
4.  For construction low-cost green agricultural production methods, the development of the mechanism for the trading of CER projects with a policy subsidy slightly lower than the construction cost can effectively promote the application of low-carbon agricultural production technologies and the development of the carbon-trading market; while the VER project trading mechanism should be adopted for the construction high-cost green agricultural production methods. If it is necessary to support the construction of projects with higher construction costs, the imposition of appropriate penalties on the agricultural sector should be considered to ensure the stability of the market.
5.  The government's reward and punishment measures have an important impact on the evolution direction and form of the carbon emission reduction project trading market. For developing countries, such as China and India, promoting the development of VER projects is a good policy tool to enhance market vitality, promote technology dissemination and popularization. For countries with more developed VER trading markets, such as the United States, policy measures should be used to promote the development of CER projects to improve the credibility and verifiability of carbon emission reductions. In addition, government subsidies for carbon-trading projects can be appropriately reduced as the market develops. Research results show that when the reduction rate is slow enough, it can reduce policy costs and improve policy efficiency without affecting effectiveness.

This study is innovative in using an evolutionary game model to analyse the feasibility of CER projects. Through simulation and prediction of the development trend, the model is able to judge the feasibility of the project. The model covers the three main actors in the promotion process of CER projects: the agricultural sector, the government sector, and carbon-emitting enterprises. It has good versatility and can analyse the stability of expected evolutionary equilibrium solutions by introducing corresponding project parameters. Based on the evolutionary results, the feasibility and sustainability of the project can be determined. Additionally, the model can be adjusted appropriately to evaluate the feasibility and sustainability of more carbon offset projects, providing a theoretical basis for policy formulation and adjustment.

**Author Contributions:** Software, formal analysis, visualization, writing—original draft preparation, H.J.; formal analysis, visualization, L.Y.; writing—review and editing, H.L.; data curation, X.B. All authors have read and agreed to the published version of the manuscript.

**Funding:** This research was funded by National Social Science Funds of China grant number 18XJY007.

**Institutional Review Board Statement:** Not applicable.

**Informed Consent Statement:** Not applicable.

**Data Availability Statement:** Not applicable.

**Acknowledgments:** I would like to express my sincere gratitude to Xiaoya Wen for her unwavering encouragement throughout the process of writing my thesis. I would also like to thank my parents and friends for their constant support and encouragement.

**Conflicts of Interest:** The authors declare no conflict of interest.

**Abbreviations**

The following abbreviations are used in this manuscript:

CDM      Clean development mechanism
CER      Certified emission reductions
CCER      Chinese certified emission reductions
VER      Voluntary emission reduction

**Appendix A. Probability Integral of Strategy Selection for Stakeholders**

$$V_{A1} = \int_0^1 \int_0^{I_A^1 - C_R + P_1 + R} \frac{-I_A^1 - C_R + P_1 + R + S_C^1 \, y}{I_C^2 \, (1-y)} dy dx \quad . \tag{A1}$$

$$V_{A2} = 1 - V_{A1}. \tag{A2}$$

$$V_{B1} = \int_0^1 \int_{-\frac{C_M + C_S^1 x + C_V x + C_S^2 z + \tilde{E}_U^* x - E_U^*}{\widetilde{C_G} - C_G + E_U - \tilde{E}_U + \tilde{E}_U^* - E_U^* + I_T}}^1 \left(1 + \frac{C_M + C_S^2 z}{(C_S^1 + C_V + \tilde{E}_U^* - E_U^* + C_G z - \widetilde{C_G} z - E_U z + \tilde{E}_U z - \tilde{E}_U^* z + E_U^* z - I_T z)}\right) dy dz \tag{A3}$$

$$V_{B2} = 1 - V_{B1} \tag{A4}$$

$$V_{C1} = \int_0^1 \int_{\frac{I_A^2 - C_E^3 - C_E^2 y + C_E^3 y + P_2 y + S_C^2}{C_E^1 - C_E^2}}^1 \left(1 - \frac{I_A^2 - C_E^3}{(C_E^3 - C_E^2 + P_2 + S_C^2 - C_E^1 x + C_E^2 x)}\right) dx dz \tag{A5}$$

$$V_{C2} = 1 - V_{C1} \tag{A6}$$

**Appendix B. Assumption and Proof of Non-Negativity for $J(x)$**

To judge the monotonicity of $J(x)$, we write the derivative of $J(x)$ to $x$. The derivative of $J(x)$ with respect to $x$ is:

$$\frac{\partial J(x)}{\partial x} = C_S^1 + C_V + \tilde{E}_U^* - E_U^* + C_G z - \widetilde{C_G} z - E_U z + \tilde{E}_U z - \tilde{E}_U^* z + E_U^* z - I_T z. \tag{A7}$$

This equation is quite complex, thus we make the assumption of $\frac{E_U^*}{E_U} = \frac{\widetilde{E_U^*}}{\widetilde{E_U}} = 0.2$ to simplify the analysis. A more complex situation will be solved through simulation in the next section. Under this assumption:

$$\frac{\partial J(x)}{\partial x} = \underbrace{C_S^1 + C_V}_{\text{Cost}} \underbrace{-(0.2 + 0.8z)(E_U - \tilde{E}_U) - z(\widetilde{C_G} - C_G) - I_T z}_{\text{Profit}}. \tag{A8}$$

The first two terms represent the government's investments in maintaining the normal operation of the certification system and the carbon-trading platform, while the last three terms represent the direct economic benefits of the government in constructing such systems, including the income from the system and the cost savings for the government from their normal operation. As the certification system and carbon-trading platform often

exhibit net losses in their initial stages of construction, meaning that costs outweigh benefits, $J(x)$ is always greater than 0 $(1 > z > 0, 1 > x > 0)$.

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
