# Peer review of "Dynamic Analysis and Simulation of the Feasibility and Stability of Innovative Carbon Emission Reduction Projects Entering the Carbon-Trading Market"

_sustainability, doi:10.3390/su15086908_

Round 1

Reviewer 1 Report

I cannot deny that the subject matter of this article is important. But this article does not show any innovation compared to the existing literature in terms of the perspective of the topic, the research methodology, the findings and the guidance for real business. The points of innovation in the manuscript are also very far-fetched.

This manuscript is only a small change from the existing evolutionary model. Such changes are not sufficient for a stand-alone article. The article's extensive mathematical derivation is redundant. The model in the manuscript has not been subjected to rigorous refinement and in-depth evolution.

The discussion section of this paper is also written in an amateurish manner and does not conform to basic academic paradigms. The horizontal and vertical comparison of this paper with the relevant literature is rather lacking. It is recommended that it be rewritten.

The literature review section of the manuscript is similarly weakly written, lacking in reading and combing through the cutting-edge literature.

In addition, the manuscript has a strong Chinese overtone, like a Chinese-to-English translation using DeepL.

In short, the manuscript does not appear to be at the basic level of a well-trained academic PhD student or scholar. 

I sincerely advise him to read more high-level English literature, improve his knowledge base in mathematical economics and try writing in English.

Therefore, I had to refuse the publication of the manuscript. This is because the manuscript is really not up to the basic level of publication. I suggest that the author submits the manuscript again after careful revision and rewriting.

Reviewer 2 Report

Designing agricultural production projects with a green focus to become CER projects is appealing due to the potential to promote the adoption of eco-friendly production technologies and decrease carbon emissions through market-based incentives. To assess the viability and sustainability of such innovative CER projects, this study developed a comprehensive analytical framework using evolutionary game methods, which allows for numerical analysis or simulation. A Certified Emission Reduction (CER) is a unit of carbon credits that represents a reduction of one metric ton of carbon dioxide equivalent (CO2e) emissions. CERs are issued under the Clean Development Mechanism (CDM) of the Kyoto Protocol, an international treaty designed to combat climate change by reducing global greenhouse gas emissions. The CDM allows industrialized countries to invest in greenhouse gas reduction projects in developing countries, and receive CERs in return. These credits can then be used to meet emissions reduction targets under the Kyoto Protocol or sold on carbon markets to organizations looking to offset their own emissions. To obtain CERs, a project must undergo a rigorous validation and verification process by an independent third-party organization, which ensures that the reduction of greenhouse gas emissions is real, measurable, and verifiable.

1.       Your paper present a model for the account of the credits, and I suggest including in a previous paragraph something about techniques using vegetable approaches that can help to make more positive the general budget. In this, we can show the use of vegetables oils in production, as it was defined in Sustainability analysis of lubricant oils for minimum quantity lubrication based on their tribo-rheological performance, Journal of Cleaner Production 164, 1419-1429 and in In pursuit of sustainable cutting fluid strategy for machining Ti-6Al-4V using life cycle analysis, Sustainable Materials and Technologies 29, e00301 because practical uses can help agriculture people to extend the use. In some countries in South Americas, soy is produced for the purpose.

2.       Conclusion 5 is not well documented: which governments, using what instruments? It is not the same Europe or Asia.

3.       Ref 24 is very good, again the works usually cite technologies that help to achieve the carbon reduction, as MQL, cryo CO2, etc.

Reviewer 3 Report

This study established an analysis model based on the evolutionary game methods, and conducted simulation analysis to evaluate the feasibility and stability of innovative carbon emission reduction projects entering to the carbon trading market. This study has some innovation and reference value. However, the authors is suggested to make further modifications in the following aspects:

1.The introduction part introduces the selection background of this study and related previous studies. If the innovation and contribution of this paper can be further highlighted, it will be more conducive for readers to understand the value of this study.

2.It is suggested that the author compare and analyze the conclusions of previous studies in the conclusion part, so as to highlight the innovation of the conclusions of this study.

3.how robust is the simulation analysis by changing the input parameters?

4.Some of the math formulas and tables can be moved to appendix so that some of the main analytical and innovative content can be better presented in the paper.

5.It is suggested that the author should strictly check and modify the references, and unify the reference format according to the requirements of the journal, and check whether there are repeated references.

6.English language and style are fine/minor spell check required. It is suggested that the author can make some polish on the language.

Reviewer 4 Report

Dear authors

My comments on the manuscript are illustrated as follows:

1- I propose to the authors to use more references which have been recently published by scholars.

2- The quality of figures 5 to 9 is very low. Please replace them with high quality figures.

3- In the manuscript, the authors should refer to the figures 5 to 9 for explanation of the results of the model.  

4- The results illustrated in the manuscript should be compared with the results presented in the previous studies.

5- The future trend of studies in the field of model applied in the manuscript should be illustrated by the authors in the conclusion and policy suggestion section.

6- The software applied for solving the mathematical equations should be introduced in the manuscript.

Reviewer 5 Report

Paper is relevant and topical. Promoting green production technologies and reducing carbon emissions is the need of the hour, and this paper adds to the body of the knowledge in this respect. Research design is sound, with adequate experimentation done. Evolutionary game framework is used. The simulation results are exhaustive and conclusive. Presentation flow and illustrations are good. Policy recommendations are provided for the benefit of the society. 42 references are cited from scholarly and good journals.  There is scope to extend the research considering international settings outside of China

Round 2

Reviewer 1 Report

I have seen the author's improvement and efforts. I think young scholars should be given more opportunities.

So, I approve of the author's revision, and there are no new comments.

Reviewer 2 Report

c

Reviewer 4 Report

Dear authors

My comments on the manuscript have been appropriately conducted.